# Mapping stakeholders' relationships management in fulfilling corporate social responsibility: A study of China's construction industry

Yuqing Zhang[1]☯*, Weiyan Jiang[2]☯, Kunhui Ye[1,3]☯*, Yuanshu Liang[1]‡, Xiaowei Wang[4]‡

**1** School of Management Science and Real Estate, Chongqing University, Chongqing, China, **2** Business School, Southwest University of Political Science and Law, Chongqing, China, **3** Research Center for Construction Economics and Management, Chongqing University, Chongqing, China, **4** Department of Building and Real Estate, Hong Kong Polytechnic University, Hong Kong, China

☯ These authors contributed equally to this work.
‡ YL and XW also contributed equally to this work.
* YuqingZhang@cqu.edu.cn (YZ); Kunhui_YE@Cqu.edu.cn (KY)

**Data Availability Statement:** All relevant data are within the paper and its Supporting Information files.

## Abstract

Businesses today face a strong call for implementing corporate social responsibilities (CSR) subject to stakeholders' multiple expectations on the creation of long-term value. However, a vast majority of companies have struggled with the challenge of stakeholder management in CSR fulfillment, giving rise to a waste of enterprise resources. This study aims to map out stakeholders' relationships in CSR fulfillment to underscore stakeholder management. 253 CSR reports published by Chinese listed construction companies from 2010 to 2021 were collected and analyzed to map the stakeholders' relationships. The results show that construction companies usually stress the importance of corporate governance and workers interest, followed by construction quality and environmental preservation in implementing CSR. They adopt three types of stakeholder management approaches: coercive, cooperative, and coordinated approaches. The study presents the latest effort in managing stakeholders' relationships in the domain of CSR research. It favors construction firms to reconsider CSR activities from the stakeholder management perspective.

## 1. Introduction

In the construction section, the diverse expectations of multiple stakeholders expose companies to social environmental impact, social welfare, and fair employment [1]. There has been a shift in the focus of construction companies as reflected in a need to improve various aspects, including sustainability challenges that require a response to governmental and the public demands. Meanwhile, stakeholders who affect or are affected by construction are increasingly concerned and motivate construction companies to properly fulfill corporate social responsibility (CSR) [2–4]. Therefore, construction companies are usually advocated to consider stakeholders' interests and make due reactions along the construction process [5]. CSR concerns

**Funding:** This research work is supported by the Fundamental Research Funds for the Central Universities (NO. 2022CDJSKPT25) and the Ministry of Education of the People's Republic of China (NO. 21JHQ092), both awarded to KY. The funder had no role in study design, data collection and analysis, decision to publish, or preparation of the manuscript.

**Competing interests:** The authors have declared that no competing interests exist.

must be incorporated into companies' strategies to formalize stakeholders' requirements and expectation [6].

CSR research in construction has gained prominence as a key agenda, possibly due to the two-sided uniqueness of construction business with its high positive and negative impacts [7]. Construction activities consume a large amount of resource, produce construction wastes, pollutants, and greenhouse gas emissions, and causes multidimensional impacts on the public and physical environment [1, 8, 9]. The complexity of these impacts cannot be merely solved by advancing construction technologies and management paradigms. Instead, embedding CSR into strategic frameworks is useful for deploying organizational resources to reduce conflicts between business operations and stakeholders' concerns [10].

According to the National Bureau of Statistics of China (2022), China's construction industry has an output value of RMB 311,979.84 billion and 51,480,200 employees. However, the construction industry in China consumed 45.5% of energy and generated 50.9% of carbon emissions in 2020 [11], suggesting that the industry has become a major pollutant source. The increasing problem of unsustainability in the construction industry draws considerable attention from researchers to explore more solutions. For instance, [12] found it worthy of highlighting the importance of CSR in the management of construction quality and safety; [13] argued that quality management and customer services should be connected to the fulfillment of CSR. While practitioners' CSR awareness stays low [14, 15], progress has been made in China to embrace communication, biodiversity, ecological conservation, and employee education [16]. Unlike developed countries where climate change, ethics, and sustainable technologies are highly recognized, China's construction business is beginning to introduce stakeholder management into CSR strategies to achieve a more balanced approach with policy.

The complex and dynamic stakeholders in the construction business hint that recognizing, prioritizing, and engaging stakeholders should be a key phase of CSR practices [17]. As reported in previous studies, stakeholder analysis in CSR fulfillment should be devoted to a number of key issues such as stakeholder-based CSR frameworks (e.g. [5, 18]), stakeholder engagement (e.g. [19, 20]), stakeholder influence strategies (e.g. [17]), stakeholder relationship management (e.g. [21, 22]). Probably in line with these key issues, [23] disclose them into two categories of management tasks: 1) promoting relationships between different project participants; 2) and analyzing the impact of stakeholders arising from the existence of 'the network of relationships', as they are important in achieving construction success. [21] develop a stakeholder management performance model for enhancing construction project success, stakeholder and organizational satisfaction, while [24, 25] use social network analysis (SNA) to establish "relationship networks", highlighting that effective relationship management of stakeholders is crucial for project success and resolving CSR conflicts [26]. Besides, there are some studies on organizational attributes towards stakeholders' power, interests, and influence, but the relationships between the organizations and their stakeholders remains unclear to support improvements in CSR fulfillment [20]. In summary, notwithstanding the growth of academic examinations on the theme of "stakeholder management in CSR", fewer efforts have been made to detect the types and meanwhile the associated relationships of stakeholders in CSR practices. This departs from the reality that construction companies need the cooperation of stakeholders to fulfill their responsibilities [27].

This study aims to map the relationships between stakeholders and finds a scientific way to manage them. We first conduct text mining of CSR reports and association analysis of CSR activities. Then, we mine the associated CSR activities, map them through a stakeholder—CSR matrix framework, and use association mining to analyze the stakeholder relationships and identify their positions. The identified stakeholder relationships from the CSR perspective deepen the understanding of stakeholder relationship management. In particular, the

proposed associated stakeholder relationship enriches previous stakeholder related research on identification, communication, and dialogue. It also provides researchers and practitioners with a scientific methodology for reference, suggests ways to manage different stakeholder relationships, and improves the efficiency of CSR resources in practice.

## 2. Literature review

### 2.1 CSR activities in construction

CSR in construction differs from those in other sectors due to the need to address various paradoxes [28], focusing on maintaining responsible aspects of construction activities while eliminating irresponsible elements [4]. On the one hand, the construction industry provides considerable jobs and makes it socially responsible to support our social and economic activities [29]. On the other hand, construction activities are often criticized for their irresponsible attributes owing to the generation of considerable negative impacts on the natural environment and poor occupational health and safety records [30]. For example, CSR in construction companies has been shown to improve corporate competitiveness, financial performance, image and business reputation [14]; as well as contribute directly or indirectly to the achievement of some sustainable development goals (SDGs), such as the SDGs for food, health, education, women, water and energy [4]. With society's focus on sustainability challenges, CSR in construction is inherently integrative, with a narrower focus on environmental activities. Therefore corporations are supposed to tailor social responsibility with society's diversified needs. This growing demand often comes from governments, competitors, customers, and environmental and social pressures [31].

Construction companies often invest in various CSR initiatives, but early returns might be limited [28]. The combined cost of acquiring these resources must be less than the total revenue generated to ensure profitability. Therefore, they prioritize high-return CSR activities like quality management, customer service, resource conservation, and waste reduction. Bundling resources for CSR implementation is an effective strategy to maximize productivity [32]. This resource optimization helps make the most of CSR investments given resource constraints [33]. Considering the resource perspective, the relationship between CSR activities assists in rational resource allocation [10]. For example, pre-construction and material recycling require simultaneous control of resource consumption and pollution. Additionally, safety education and on-site training must be complemented by a safety management system to prevent accidents [34]. Based on this, Hypothesis 1 is proposed.

*Hypothesis 1 (H1)*: Companies prioritize the implementing of key CSR activities and leverage their associated relationships in the fulfillment process.

### 2.2 A stakeholder-based view on CSR

There are many views on CSR. Initially, CSR emerged as a purely ethical stance [35]. A socially responsible business should be "profitable, law-abiding, ethical, and a good corporate citizen" [36] and "doing better by doing good" [37]. Two main schools of thought point to CSR. On the one hand, company's principal responsibility is to maximize its profits within the boundaries of the law, i.e. the "shareholder supremacy" view [38]. [39] states that shareholder value is the only value that could be maximized and argues that pursuing social goals dilutes enterprises' main goals, increases costs, and reduces economic efficiency, competitiveness and profitability.

On the other hand, a stakeholder-based view has been presented to emphasize individuals and groups of "stakeholders", who can affect or are affected by the achievement of the company's objectives' [40]. [41] argue that businesses develop positive relationships with society,

communities and their stakeholders to help become better citizens. The World Business Council for Sustainable Development also states that CSR reflects a company's commitment to its employees and their families, the local communities and society [42]. In other words, CSR is a holistic relationship between a company and all its stakeholders, which creates shared values for the organization and its stakeholders. Therefore, CSR can be treated as a company's stakeholders' ethical or responsible treatment.

Stakeholder-based view provides a useful instrument for CSR fulfillment. Researchers have widely adopted it to investigate key aspects of CSR activities rooted in economic, environmental and social dimensions [43]. CSR activities serve as a means for companies to nurture relationships with various stakeholders [44], and CSR activities' prioritization and associated relationships result from continuous interactions with various stakeholders. Companies respond to pressure from stakeholder groups by changing their CSR relationships. While the stakeholder-based view emphasizes the importance of all stakeholders, achieving a balance can be challenging [45]. Stakeholder salience, based on factors like power, legitimacy, and urgency, helps organizations identify which stakeholders' interests are most crucial and sets their priorities [46]. This context leads to the proposal of Hypothesis 2.

*Hypothesis 2 (H₂)*: Companies prioritize meeting the needs of key stakeholders.

## 2.3 CSR stakeholder relationship in construction

Analyzing and managing stakeholder relations demands for time and enterprise resources in reaction to stakeholders' requirements and expectation [47]. The analysis and management is supposed to build on the development and implementation of organizational policies and practices that consider all stakeholders' goals and concerns [48]. Managing the relationship with each stakeholder should also focus on developing and strengthening the organization's connection with each stakeholder, driven by organizational behavior and creativity [49].

Stakeholder analysis, a part of the stakeholder management process [50], involves companies gathering information on stakeholder issues to manage changing interests [51]. Institutions that shape incentives and legitimacy affect the social behavior of companies. Stakeholders gain legitimacy and power from these institutions, impacting the company's CSR strategy. Additionally, legitimacy theory presumes a social contract between organizations and society [52]. In the context of CSR, supporting effective tools to convey the organization's legitimizing actions and gaining legitimacy is considered to be an important resource for the organization [53]. "Stakeholder legitimacy" is defined when stakeholders claim legitimacy in organizations, addressing their demands [54]. This interplay between legitimacy and the stakeholder perspective fosters dialogue between organizations and stakeholders [55]. Leveraging legitimacy and understanding stakeholder salience helps organizations manage multi-stakeholder relationships based on the priority given to stakeholder advocacy.

Implementing stakeholder relationship management throughout the organization has an important role in CSR management [56]. It centers on creating, maintaining, and coordinating stakeholder relationships to enhance value and prevent ethical issues. CSR stakeholder relationships are managed due to the complementary nature of CSR activities. Shared resource requirements strengthen stakeholder association, promoting collaboration based on complementary capabilities and efficient resource use.

In construction, CSR is a means of gaining legitimacy [29]. Managing stakeholder relationships offers the potential for addressing complex environmental issues specific to construction [17]. Contracts enable construction companies to engage in socially responsible activities, gaining social recognition within legal boundaries [30]. Legality drives construction companies to emphasize coordinated relationships, especially optimal resource allocation amid

limitations and conflicting stakeholder interests. Real-world cases reveal that CSR often involves collaboration with other organizations. Many CSR activities, addressing meta-issues like community poverty and environmental pollution, require collaboration with various organizations with specific expertise [27]. Companies can view certain stakeholder groups holistically, sharing resources and power across organizations, enhancing collaboration among multiple stakeholders. These relationships enable fulfilling multiple CSR objectives, facilitating decision-making within resource constraints. In summary, Hypothesis 3 is proposed.

*Hypothesis 3 (H₃)*: There are associated relationships between stakeholders.

## 3. Research methods

Three steps are adopted to actualize the research objectives. First, we conduct a systematic literature review to identify the CSR activities and use text mining of the collected reports; then employ an association analysis of CSR activities. Second, we segment stakeholders corresponding to each CSR activity based on samples from the Engineering News Record (ENR) Chinese top 5 and related literature to establish a stakeholder—CSR matrix framework (S-CSR-M). Third, the associated CSR activities obtained in the first step are mapped to stakeholders through S-CSR-M; then the stakeholder relationships are obtained through association mining. The research methodology is outlined in Fig 1.

### 3.1 Text mining of CSR activities

We conduct a systematic literature review of CSR in the construction industry based on the Prisma checklist of [57]. We search the Web of Science and Scopus databases using the Boolean search rule: ((("corporate social responsibility" OR "CSR") AND ("construction" OR "contractors")) to select journal papers as of May 2023. We then screen 128 highly relevant papers to this study based on the content and topics, and construct a framework for CSR activities based on 38 of them, as shown in S1 and S2 Tables.

The research proposes a text mining approach to analyze CSR reports, including text cleaning (like removing stop words), training a word vector model of skip-gram in Word2Vec,

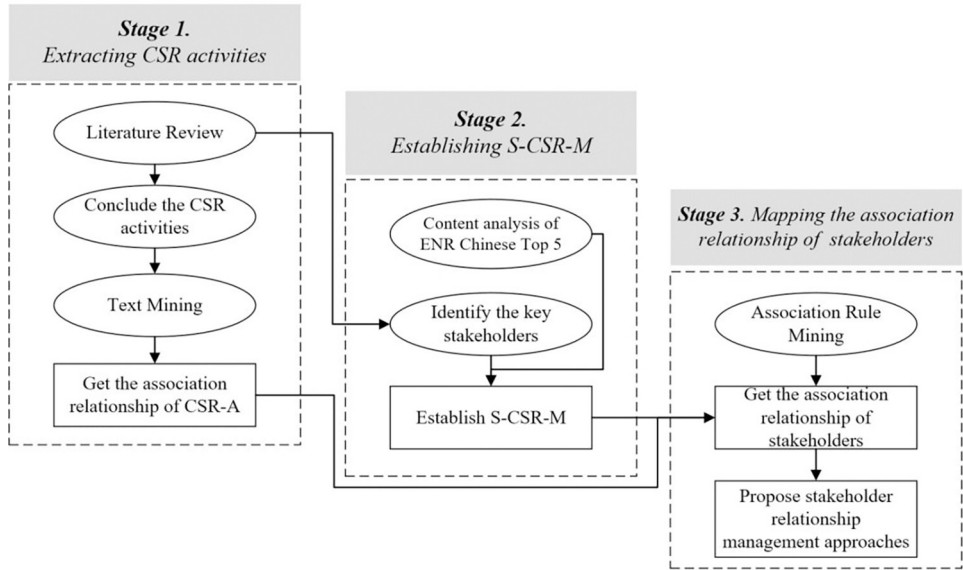

**Fig 1. The research flow chart.**

**Table 1. Reliability results of text mining.**

| Type | (Actual value, Predicted value) | Numbers | Reliability Index | Equation | Results |
|---|---|---|---|---|---|
| True Positive (TP) | (1,1) | 1309 | Accuracy | $\frac{TP+TN}{All\ sample}$ | 89.42% |
| False Positive (FP) | (0,1) | 283 | Precision | $\frac{TP}{TP+FP}$ | 82.22% |
| True Negative (FN) | (1,0) | 156 | Recall | $\frac{TP}{TP+FN}$ | 89.35% |
| False Negative (TN) | (0,0) | 2402 | F1-score | $2 \times \frac{Precision \times Recall}{Precision + Recall}$ | 0.856 |

constructing a CSR-activity model keyword database by using the *most_similar* method, quantitating results to the value of 0 or 1, and verifying text mining accuracy. We randomly select 50 CSR reports according to the annual number to ensure reliable results, and compare text mining results with manual coding. The value of accuracy, precision, recall and F1-score is calculated by examining the true positives (TP), false positives (FP), true negatives (TN) and false negatives (FN), as shown in Table 1. F1-score is found to be 0.856, indicating the text mining approach is reliable.

## 3.2 Establishing a stakeholder—CSR matrix framework

We use a modified version of the linear responsibility diagram, originally developed by [18], to identify stakeholder expectations and consider CSR in a structured way. The linear responsibility diagram is commonly used in management and allows for identifying people, issues, and relationships between them. The matrix allows organizations to match their own CSR activities with the ethical values of their stakeholders, and identify CSR issues desired by each stakeholder. By answering key questions posed by [36, 58] of "whom we are accountable to" and "what do they need", the tool can help companies understand who they are accountable to and what they need. This approach integrates stakeholders, their ethical responsibilities, and their relationships. It has been applied to a major UK contractor to understand stakeholder values and commonalities [18].

Referring to [5], we argue that the core stakeholders of construction companies in the context of CSR include government, shareholders, employees, suppliers, end users, community groups, environment and resources agencies (E&R), partners (such as designers, consulting, sub-contractor, operators, industry association), supervisor, competitors, union, minority, NGOs and media. The S-CSR-M identifies stakeholders and their ethical interests, allowing organizations to map and identify consultations on support, importance, partners, and ethical responsibilities. The vertical axis includes CSR topics and activities derived from literature and ENR Chinese top 5 CSR reports (see S3 Table). The horizontal axis identifies stakeholders. Once the fundamental matrix is established, the CSR reports of ENR Top 5 Chinese construction of 2010, 2015, and 2020 are collected and sorted according to the key stakeholders and CSR activities. Finally, the S-CSR-M is further refined through team discussions, see the Table 2 and S4 Table. The 83 CSR activities correspond to 13 stakeholders and the CSR activities of each stakeholder are counted.

## 3.3 Association analysis of CSR activities and stakeholders

Association rule mining (ARM) is an unsupervised machine learning (ML) method used to discover hidden relationships in transactions [63]. It demonstrates knowledge of interrelationships between items in a dataset and is a main tool of knowledge discovery data (KDD). Such rules demonstrate knowledge of the interrelationships between items in a dataset [64]. The

**Table 2. The stakeholder—CSR matrix (S-CSR-M).**

| CSR codes | Government | Shareholders | Employees | Suppliers | End users | Community group | E&R | Partners | Supervisor | Competitors | Union | Minority | NGOs and Media |
|---|---|---|---|---|---|---|---|---|---|---|---|---|---|
| R01 | √ | | | | | | | | | | | | |
| . . . | | | | | | | | | | | | | |
| R11 | √ | √ | | | | | | | | | | | √ |
| G01 | √ | | √ | | | | | | | | | | |
| . . . | | | | | | | | | | | | | |
| G08 | √ | | | | | | | | √ | | | | |
| Q01 | √ | | | | √ | | | √ | | | | | |
| . . . | | | | | | | | | | | | | |
| Q15 | √ | | | | √ | | | | | | | | |
| P01 | √ | | | √ | | | | | | | | | |
| . . . | | | | | | | | | | | | | |
| P07 | | | | √ | | | √ | | | | | | |
| W01 | | | √ | | | | | | | | √ | | |
| . . . | | | | | | | | | | | | | |
| W14 | | | √ | | | | | | | | | | |
| C01 | | | | | | √ | | | | | | | |
| . . . | | | | | | | | | | | | | |
| C14 | | | | | √ | | | | | | √ | √ | |
| E01 | | | | | | | | | | | | | |
| . . . | | | | | | | | | | | | | |
| E14 | √ | | | | | √ | √ | | | | | | √ |
| *Total No.* | *46* | *11* | *17* | *5* | *13* | *20* | *15* | *22* | *14* | *14* | *14* | *7* | *27* |

Note: Based on [5, 12, 59–62], and the reports of "ENR's 2022 Top 10 Chinese Contractors" in 2010, 2015, 2020. R: management of responsibility; G: corporate governance; Q: safe construction and quality; P: good partnership; W: workers interest; C: the well-being of local communities; E: environment preservation. Please see S4 Table for the full table.

term "transaction" refers to a group of items with an implicit or explicit relationship to build a meaningful collection together. Social network analysis (SNA) is also often used to map the interrelationships between stakeholders and social behaviors (e.g. [23, 65]). However, SNA may have limitations in boundary, bias and uncertainty [23]. ARM is increasingly applied in construction management research (e.g. [66–68]). Appling ARM to the relationship of CSR activities and translates it into a collection of transactions consisting of stakeholder objects. The process is illustrated in Fig 2.

Let a rule be defined as: ($X{\rightarrow}Y$), where $X$, $Y{\subseteq}I$ and $X{\cap}Y = \emptyset$. Here $I = \{i_1, i_2,. . .,i_n\}$ is the set of $n$ CSR activities, $X$ is called the antecedent (or LHS) and $Y$ is called consequent (or RHS). An association rule is a rule that satisfies the minimum *support* (*min_sup*) and minimum *confidence* (*min_conf*) threshold requirements. *Support* and *confidence* are two of the most basic and critical metrics used in ARM to quantify the strength of a rule. Associations that satisfy *min_sup* and *min_conf*, which are pre-specified by the user, are classified as strong associations. After testing, a *min_sup* of 0.7 and *min_conf* of 0.9 are chosen. The strength of association rules is evaluated using *lift*, *leverage*, and *conviction* [69].

$$\text{Lift}(X \rightarrow Y) = \frac{\text{confidence}(X \rightarrow Y)}{\text{support}(Y)} \tag{1}$$

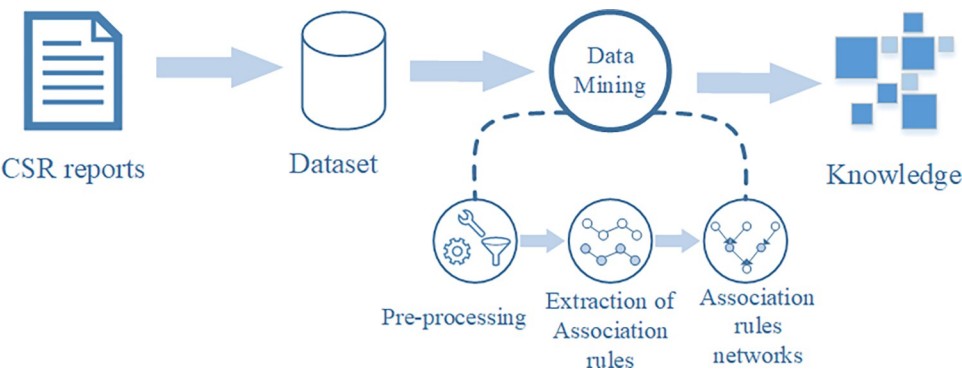

**Fig 2. The flowchart of KDD.**

*Lift* is the ratio of "the proportion of transactions containing X that also contain Y" to "the proportion of transactions containing Y". The *lift* reflects the correlation between X and Y in the association rule, with higher values indicating a higher positive correlation. Where *support* is the frequency of item combinations in the dataset, i.e., how often itemsets X and Y appear together in a transaction. *Confidence* is the frequency with which item set X appears in a transaction containing both X and Y. See the equations:

$$\text{Support} = P(X \cap Y) \tag{2}$$

$$\text{Confidence}(X \rightarrow Y) = \frac{\text{support}(X \rightarrow Y)}{\text{support}(X)} \tag{3}$$

In addition, the quality of association rules is assessed using *leverage* and *conviction*. *Leverage* indicates how much more often than antecedent and consequent occur together when they are independently distributed. *Conviction* is the probability of X occurring without Y occurring in terms of the difference, i.e., the probability that the rule' prediction is wrong. The larger the value, the more associated X and Y are. The equations are as follows:

$$\text{Leverage}(X \rightarrow Y) = \text{support}(X \rightarrow Y) - \text{support}(X)\text{support}(Y) \tag{4}$$

$$\text{Conviction}(X \rightarrow Y) = \frac{1 - \text{support}(Y)}{1 - \text{confidence}(X \rightarrow Y)} \tag{5}$$

To assess validity and measure selection, correlation coefficient, cosine, Jaccard, Klosgen, and Loevinger values are calculated for each rule, also shown in Table 3.

## 4. Data analysis and results

### 4.1 Materials and data

In the construction industry, CSR reporting has become increasingly important as stakeholders become more aware of the impacts of construction projects on economic, social, and ecological aspects [70]. This study collects CSR reports, sustainability reports, and ESG (environmental, social and governance) reports of Chinese listed construction companies from 2010 to 2021, totaling 253 reports. The sources of the report include official websites of construction enterprises, and Shanghai Stock Exchange (*http://www.sse.com.cn/*) and Shenzhen Stock Exchange (*http://www.szse.cn/*).

**Table 3. Measures to evaluate the strength of association rules.**

| Measures | Definition | Equation |
|---|---|---|
| Correlation Coefficient | The covariance between two items divided by their standard deviation. A value of 0 indicates independence. Also known as Phi correlation. (range [−1,1]) | $\frac{supp(X \to Y) - supp(X)*supp(X)}{\sqrt{supp(X)supp(Y)(1-supp(X))(1-supp(Y))}}$ |
| Cosine | Modified cosine similarity metric for ARM. A value of 0.5 means no correlation. (range [0,1]) | $\frac{supp(X \to Y)}{\sqrt{supp(X)supp(Y)}}$ |
| Jaccard | Similarity between two sets of transactions that contain the items in X and Y. (range [0,1]) | $\frac{supp(X \to Y)}{supp(X)+supp(Y)} - supp(X \to Y)$ |
| Klosgen | Test of independence between the antecedent and the consequent. A value of 0 indicates independence. (range [−1,1]) | $\sqrt{supp(X \to Y)}*(conf(X \to Y) - supp(Y))$ |
| Loevinger | The probability that Y is in a transaction containing X. The value of 0 indicates independence. Also known as the Certainty Factor (CF). (range [−1,1]) | $\frac{conf(X \to Y) - supp(Y)}{1 - supp(Y)}$ |

Note: *Supp* is support, and *conf* is confidence.

Figs 3 and 4 illustrates the basic situation of the reports. The number and percentage of reports published by listed construction companies each year are shown in Fig 3, showing an increasing trend, indicating that more and more construction companies are starting or consistently publishing independent CSR information. The lowest is 12 reports in 2010, accounting for about 33.33%, the highest is 26 (47.27%) in 2019. Meanwhile, Fig 4 shows the proportion of provinces, with the most companies publishing in Beijing, accounting for about 49.01%, and less in Guangdong (3.16%). In addition, 91.30% of the companies that issued reports were SOEs (state owned enterprises).

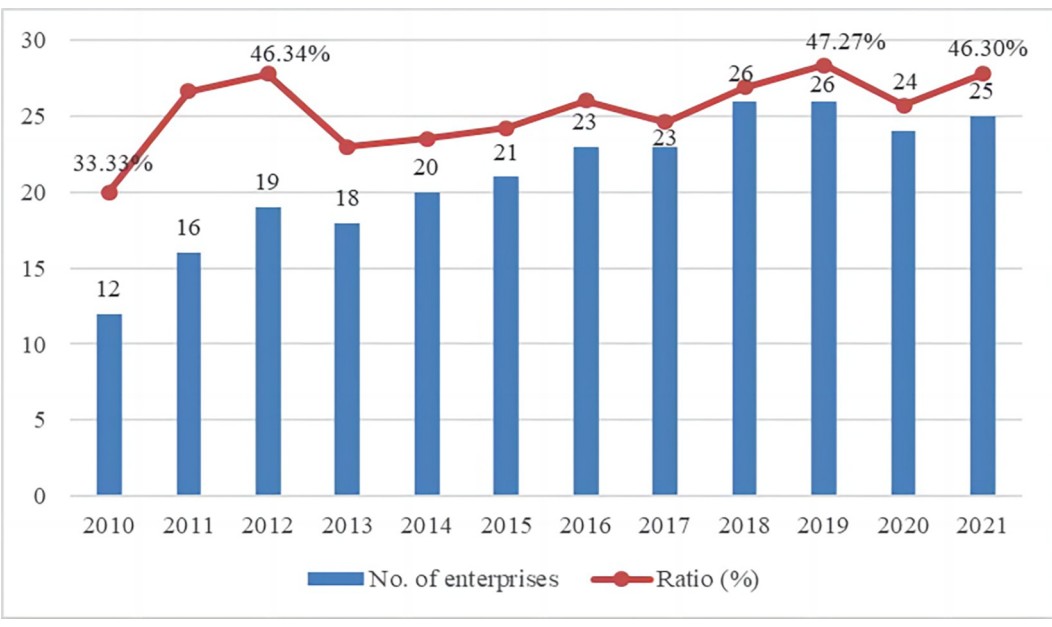

**Fig 3. Number and proportion of CSR reports issued by listed construction companies 2010–2021.**

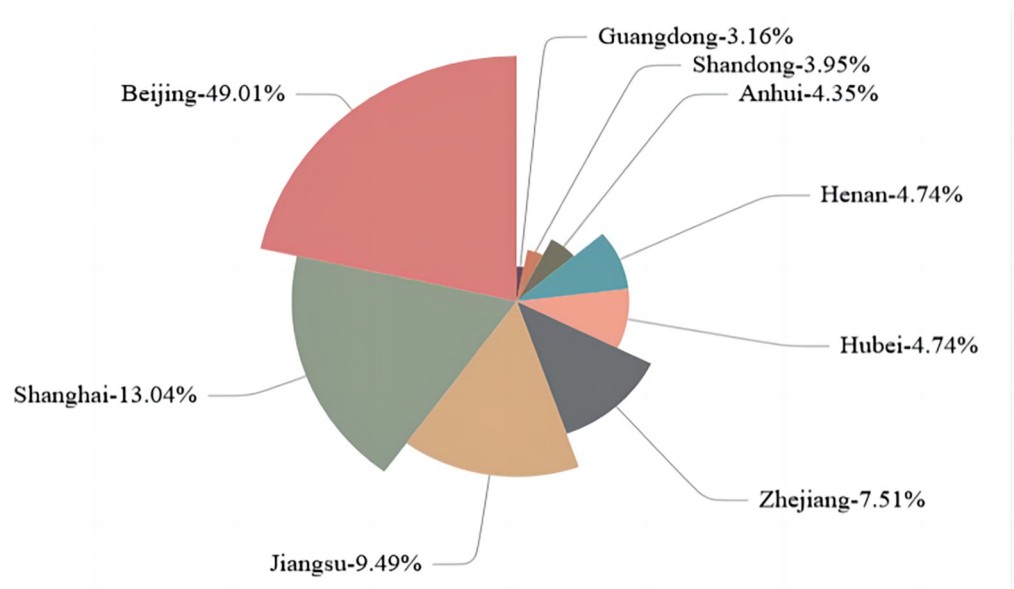

**Fig 4. Proportion of CSR reports issued by listed construction companies by province 2010–2021.**

## 4.2 Conducting association analysis on CSR activities

Association rule generation algorithms include Apriori, Eclat, and FP-Growth, with FP-Growth and Eclat being suitable for large datasets, while Apriori is used in this study due to due to its conciseness and ease of understanding in small datasets [71]. The Apriori algorithm is implemented in Python 3.7.6 and R 3.6.3.

After generating the frequent itemsets, we use the metrics reported in 3.3 and Table 3 to calculate the strength of itemsets. It is also assumed that the strength of $X \rightarrow Y$ may not be equal to the strength of $Y \rightarrow X$, since some metrics are not bidirectional. The selected a priori rules set in this study generate a range of *support* (0.7–1.0) and *confidence* (0.9–1.0) with *lift* greater than 1, totally 2893 rules. Table 4 is a fragment of the quasi-extracted rules and their values.

In Fig 5, the source indicates the antecedent, the arrow indicates the direction the relationship points to, the circle in the middle indicates the size of the *confidence* level of this rule, the

**Table 4. Snippets of association rules for select CSR activities.**

| antecedents | consequents | support X | support Y | support X→Y | confidence | lift |
|---|---|---|---|---|---|---|
| R03 | G05 | 0.972 | 0.976 | 0.953 | 0.980 | 1.003 |
| E04 | G05 | 0.941 | 0.976 | 0.925 | 0.983 | 1.007 |
| G01 | R03 | 0.945 | 0.972 | 0.925 | 0.979 | 1.007 |
| G01, G05 | R03 | 0.921 | 0.972 | 0.905 | 0.983 | 1.011 |
| G01 | R03, G05 | 0.945 | 0.953 | 0.905 | 0.958 | 1.006 |
| leverage | conviction | Correlation Coefficient | cos | Jaccard | Klosgen | Loevinger |
| 0.003 | 1.167 | 0.132 | 0.978 | 0.956 | 0.003 | 0.143 |
| 0.006 | 1.411 | 0.181 | 0.965 | 0.932 | 0.007 | 0.291 |
| 0.006 | 1.323 | 0.170 | 0.965 | 0.932 | 0.006 | 0.244 |
| 0.010 | 1.612 | 0.219 | 0.957 | 0.916 | 0.010 | 0.380 |
| 0.005 | 1.134 | 0.109 | 0.954 | 0.912 | 0.005 | 0.118 |

Note: The length of frequent item-sets varied between 1 and 6 (those with *support* > = 0.70, *confidence* > = 0.90). This dataset includes 2893 rules; this study did not assume bidirectionality in the rules. Due to space limitations, this table presents only several rules.

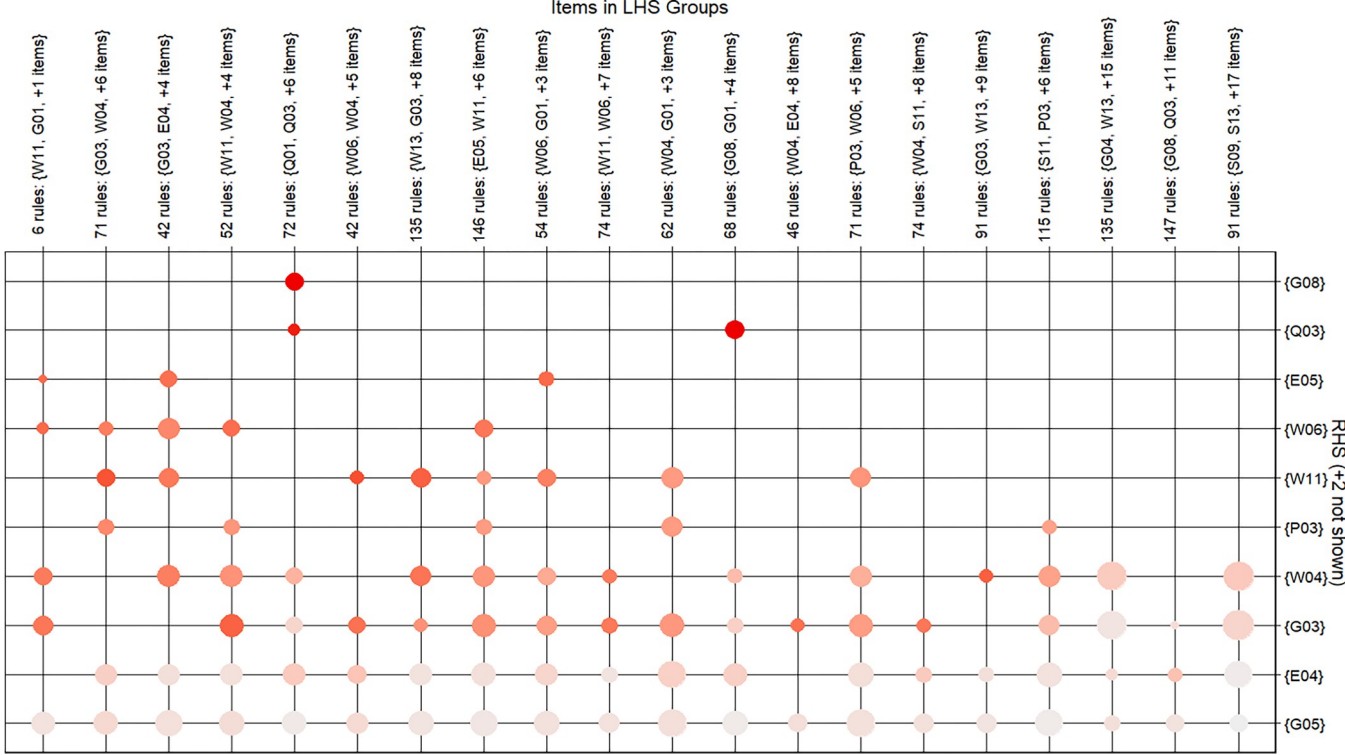

**Fig 5. Graph of associated CSR activities.**

depth of the circle color indicates the size of the *lift*, and the end of the arrow pointing to the consequent of the rule. It is a causal graph of CSR activity association rules, and the graph shows the relationship of frequent item sets such as {R03}, {G01}, {G03}, {G05}, {G08}, {P03}, {S11}, {W04}, {W06}, {W11}, {W13}, {Q01}, {Q03}, {E04}, {E05} as antecedent, consequent or intermediate nodes. They are mostly CSR activities such as corporate governance and workers interest, followed by construction quality and environment preservation.

Fig 6 shows the relationship between the LHS and RHS of the rules in the grouping matrix. The balloons in the graph of the grouping matrix indicate the level of associated interest in the group, the *lift* is the color shade of the circles, and the size of the circles indicates the size of the *support*, and a certain result (RHS) in a given prior item (LHS). The *lift* decreases from the upper left corner to the lower right corner. The CSR activities with the strongest correlation (darkest dot color) according to the *lift* parameters are {Q01, Q03} with {G08}, {G08, G01} and {Q03}.

## 4.3 Analyzing stakeholders in S-CSR-2M

The matrix is a mapping tool that allows to identify ethical responsibilities together with stakeholders. Although its approach is linear, it allows stakeholders and CSR to be combined in a simple format so that an organization can match its own ethical values with those of its stakeholders. Combined with the CSR activities corresponding to each type of stakeholder in S-CSR-M in the attached table, calculate the frequency of each type of stakeholder through the Eq (6), see the S4 Table.

$$FrequencyS - CSR_i = \frac{\sum_1^7 S - CSR_i}{\sum_1^7 S - CSR_1 + \sum_1^7 S - CSR_2 + \cdots + \sum_1^7 S - CSR_{13}} *100\% \quad (6)$$

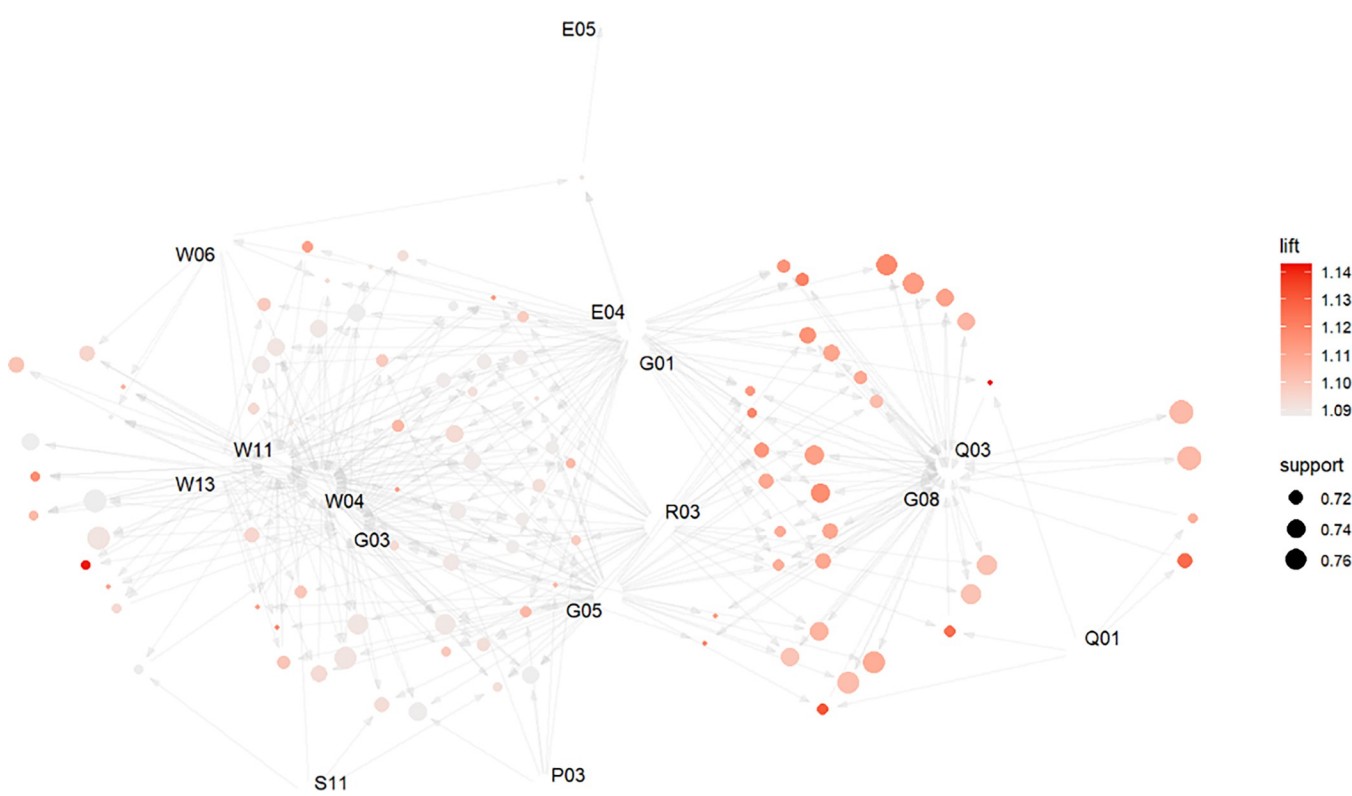

**Fig 6. A grouped matrix plot of association rules for CSR activities.**

Fig 7 shows the mapping of 13 types of stakeholders to each CSR activity. The government is the stakeholder who pays the most attention to the implementation of various CSR activities in construction enterprises, paying attention to more than half of the CSR activities (46/83), and is also the most concerned among all stakeholders (about 21.00%). Followed by NGOs and Media (12.33%) related to various CSR activities of construction companies, especially issues related to the environment. Then there are partners (10.05%) on CSR activities related to construction quality. In addition, stakeholders such as E&R agencies, union, and employees will be more mapped to relevant CSR categories.

## 4.4 Mapping associated stakeholder relationships management

Table 5 shows the stakeholders mapped by the left and right CSR activities in all the association rules that meet the conditions. The previous CSR activities involve more stakeholders, most of whom are government (13.55%), NGO and media (12.06%), shareholders (11.12%), union (10.44%) and other stakeholders. The least involved minority (1.03%) and community group (1.04%). The results obtained in the latter item also show that the government (15.45%), shareholders (15.70%), and NGO and media (13.72%) are the most mapped stakeholders. Under this condition, no community group and minority appear in the latter item of the key rule. At the same time, by comparing the stakeholders before and after the items in Table 5 (column 6), stakeholders such as employees (+4.22%), union (+3.33%), and suppliers (+1.98%) are in the frequency of the left rules is higher, indicating that they will be used as the antecedents. While stakeholders such as NGOs and media (-5.47%), community group (-3.92%), E&R agencies (-2.34%), suppliers (-2.03%) appear more frequently in the latter rules, indicating that they tend to appear after the former stakeholders.

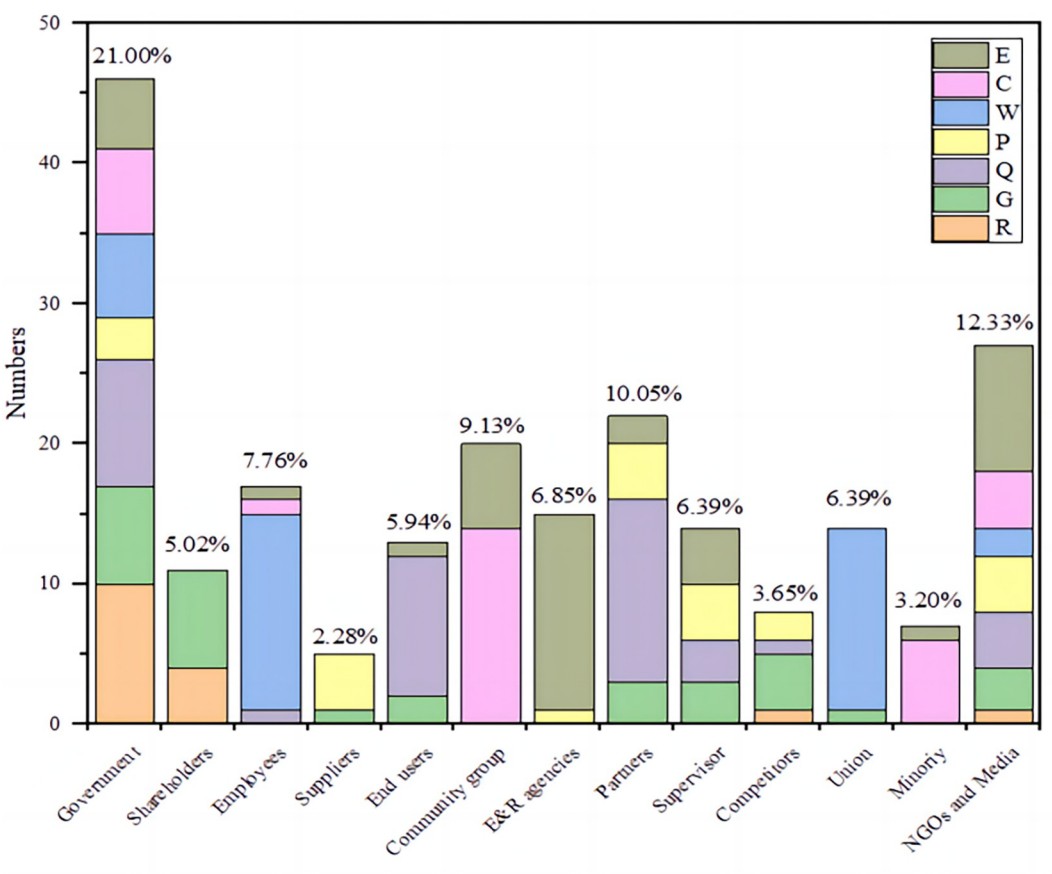

**Fig 7. CSR activities for each type of stakeholder in S-CSR-M.**

**Table 5. Numbers of antecedents and consequents stakeholders.**

| Stakeholders | antecedents No. | Ratio 1 (%) | consequents No. | Ratio 2 (%) | Ratio 1–2 (%) |
|---|---|---|---|---|---|
| Government | 2834 | 13.55 | 2273 | 15.45 | -1.90 |
| Shareholders | 2326 | 11.12 | 2311 | 15.70 | -4.59 |
| Employees | 1782 | 8.52 | 632 | 4.29 | +4.22 |
| Suppliers | 478 | 2.28 | 45 | 0.31 | +1.98 |
| End users | 1638 | 7.83 | 1393 | 9.47 | -1.64 |
| Community group | 217 | 1.04 | 0 | 0 | +1.04 |
| E&R agencies | 1172 | 5.60 | 719 | 4.89 | +0.72 |
| Partners | 1873 | 8.95 | 1429 | 9.71 | -0.76 |
| Supervisor | 1944 | 9.29 | 1425 | 9.68 | -0.39 |
| Competitors | 1734 | 8.29 | 1424 | 9.68 | -1.39 |
| Union | 2184 | 10.44 | 1046 | 7.11 | +3.33 |
| Minority | 215 | 1.03 | 0 | 0 | +1.03 |
| NGOs and Media | 2524 | 12.06 | 2019 | 13.72 | -1.66 |
| Total No. | 20921 | 100 | 14716 | 100 | - |

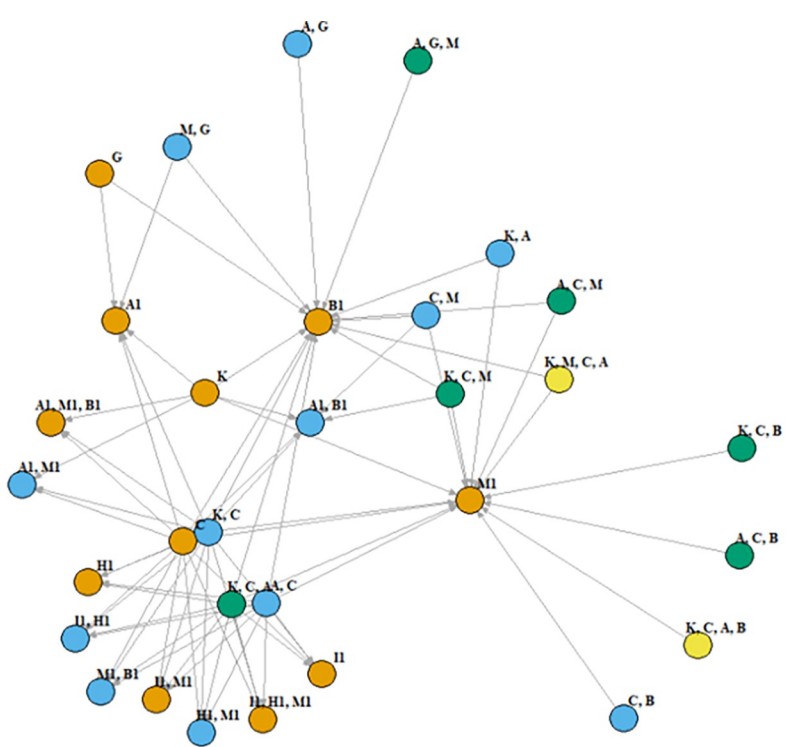

**Fig 8. The map of association relationship for stakeholders.**

Based on the above-mentioned stakeholder rules related to CSR activities, with given context items, the secondary association mining of the group relationship is performed on the mapped stakeholders. For example, when $CSR_A \rightarrow CSR_B$, the stakeholder relationship under the S-CSR-M model forms a contingency table of $\{ST_A \rightarrow ST_C\}$ before and after, and the number of stakeholders in the former and the latter is indeterminate. According to the conditions *min_support* 0.3, *min_confidence* 0.5, *lift* > 1.0, and delete the rules with duplicate stakeholders before and after the item. Finally, 72 rules with a given direction are screened, see S4 Table. These rules are based on the association relationship between stakeholders in CSR activities. The length of itemsets ranges from 2 to 6, and they also show a combination mode. Through the visualization of the stakeholder relationship as shown in Figs 8 and 9, it shows the relations and paths of different stakeholders. Paths ① and ② represent simple association relations, i.e., LHS to RHS, and path ③ shows the relations of intermediate nodes. The result in the combined mode is similar, the preceding item or the following item is the combination of the preceding and following items. Meanwhile, Fig 9 shows that the former item (LHS) covers employees (C), environment and resources agencies (G), union (K); while the latter item (RHS) includes partners (H1), and supervisor (I1); and government (A/A1), shareholders (B/B1), NGOs and media (M/M1) appear in the LHS and RHS.

## 5. Findings and discussion

### 5.1 Associations of CSR activities

Figs 5 and 6 show that construction companies prioritize their CSR efforts in areas such as corporate governance, worker welfare, safe construction practices, and environmental preservation. They address both mandatory requirements and market-based initiatives, including enhancing corporate governance structure, legal compliance, information disclosure, ensuring

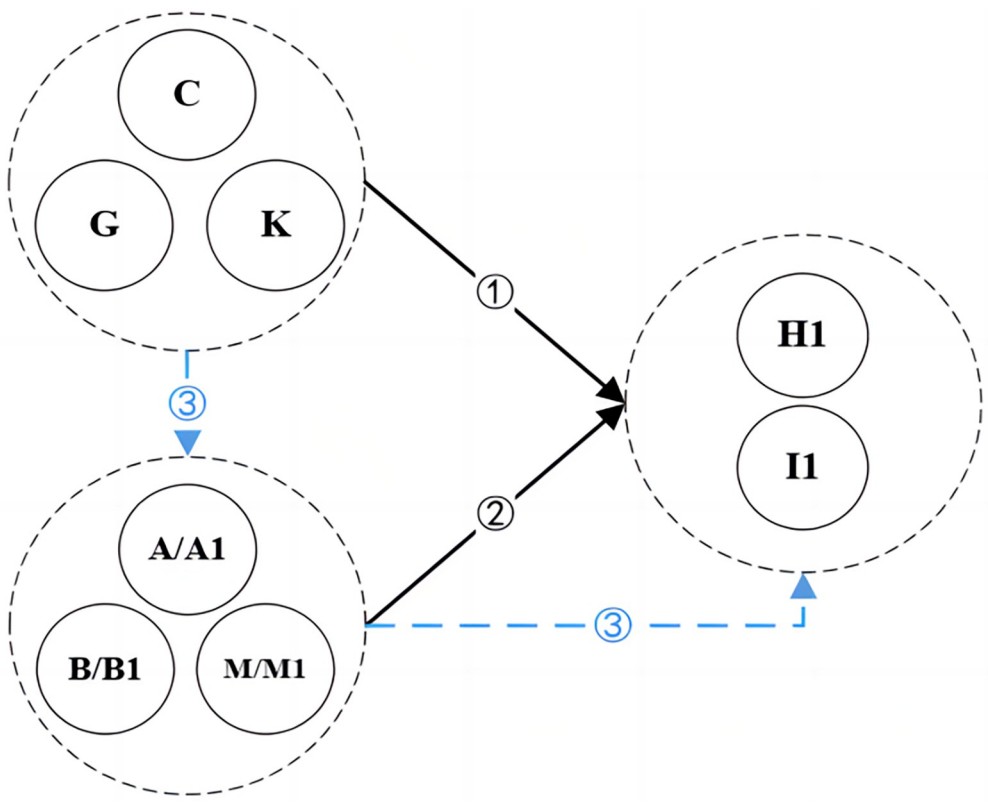

**Fig 9. Direction of association map for three types of stakeholders.**

labor contracts, promoting equal employment, and more. Corporate governance, an integral part of social responsibility standards like ISO26000 and the "China Social Responsibility Report Compilation Guide", enables resource allocation, stakeholder coordination, and facilitates CSR implementation and information sharing [14, 72]. Additionally, companies focus on improving employee well-being, encompassing health and safety, while fostering a fair and healthy work environment. These measures aim to reduce employee concerns, increase job satisfaction, and drive value creation [5]. Aligning with [73], this study supports the significance of construction quality and environmental protection as essential CSR dimensions in the Chinese construction sector. Differentiating from competitors through establishing robust R&D and innovation systems is a crucial factor for development [74]. Meeting environmental protection requirements, influenced by government regulations and industry standards, is also a pressing ethical responsibility. Chinese contractors exhibit a greater emphasis on implementing recycling systems and pollution emission control measures to mitigate negative environmental factors such as noise and dust, aligning with previous research (e.g. [15, 34, 75]).

CSR implementation is not isolated or scattered, but associated through a combined method, and integrates social, environmental, and economic factors into corporate strategy and practices, resulting in cost reductions, enhanced innovation, and strengthened environmental protection [76]. This balanced approach fosters long-term sustainability by considering the values associated with social, environmental, and economic aspects [77], which confirms $H_1$. Because CSR consumes resources, construction companies typically expand their resources to implement one or more CSR activities. In order to generate profit, the sum of the costs of acquiring these resources must be less than the total revenue that these resources

together generate. Under the condition of given resources, CSR activities are linked to limited capabilities and resources, and information such as the cost of implementing CSR and the scarcity of resources are carefully analyzed. Selective CSR activities not only create CSR differentiation and competitiveness, but also generate mutual benefits for stakeholders. To optimize resource utilization, enterprises should adopt a combined CSR performance approach, as supported by existing research and depicted in Figs 5 and 6. For example, when an enterprise serves the community, it can combine community greening, caring, publicity and other activities to reduce the number of repetitions and the use of resources; enterprises can reduce emissions and produce more environmentally friendly products due to consumers' environmental demands.

## 5.2 Key stakeholder involved in construction

Table 2 and Fig 7 demonstrate the mapping of relationships between 83 CSR activities and 13 stakeholder groups, providing insight into the preferences of various stakeholders towards CSR activities of Chinese construction companies, and confirming $H_2$. As it shows, governments, NGOs and media, and partners are most stressed in the construction companies. Governments, NGOs and media organizations are increasingly pressuring construction businesses to manage their performance based on new social needs driven by SDGs [5]. While governments and construction companies may not have a direct contractual relationship, governments exert significant influence through laws and regulations, and companies are encouraged to utilize CSR to address social and environmental issues [57, 78]. Compared with other countries, Chinese construction companies have a unique governance model in which the Chinese government has greater control over large, state-owned construction companies. The management of these companies is appointed by the government and has major decision-making power in the company. Therefore, the influence of government on CSR has been highlighted.

The main impact of NGOs is to use their expertise to make appeals to persuade governments or companies to respond to social responsibility issues. In line with the findings of [79], this study also states that NGOs place the most importance on issues related to the environment and sustainable development, as it shows in Fig 7. The media therefore have a large role in developing countries as they greatly influence information disclosure and increase corporate transparency through exposure. It promotes strong partnerships with stakeholders and facilitates knowledge exchange and sharing both internally and externally. Meanwhile, partners primarily prioritize the economic viability of the organizations they supply to ensure adequate remuneration, and take building quality CSR activities as more significant considerations. Responsible sourcing is often considered an integral part of CSR. Construction contractors select responsible partners and document and evaluate partner' CSR commitments and performance, both as a strategy for implementing CSR practices and as a strategy for communicating corporate benefits.

## 5.3 Strategies for stakeholder relationship management

The findings from Figs 8 and 9 categorize stakeholders into three types, pre-stakeholders, intermediate stakeholders, and post-stakeholders, each corresponding to distinct stakeholder relationship paths; consistent with $H_3$. First, pre-stakeholders like employees, environmental agencies, and unions wield significant influence across the project life cycle, embedding critical CSR aspects like human resource policies, human rights, welfare, and environmental protection [60]. Companies worldwide have started promoting employee engagement in environmental actions to enhance CSR. Second, post-stakeholders encompass partners and supervisors who tackle issues exceeding individual organizations capacities. Collaborations

with these stakeholders contribute specialized resources and knowledge, creating a coordinated front in the construction industry [27]. The value of their resources determines demand fulfillment. Their resources' importance to the company's viability determines the level of demand met. Third, intermediate stakeholders involve government, shareholders, NGOs and media. They provide internal and external oversight, enhancing decision-making efficiency, legitimacy, and resource consolidation [34]. Government involvement often responds to regulatory requirements, while NGOs and media act as resource consolidators and relationship facilitators. In contrast, the CSR outcome relationship is stronger when stakeholders have greater power and legitimacy and when regulation increases. For example, political legitimacy is also a strategic resource for companies, and the government, as a more influential stakeholder, provides more ways to obtain public resources.

On the one hand, clearly identifying the stakeholders and their types can avoid the misallocation of resources to non-stakeholder groups or stakeholders with no legitimate interests or concerns. On the other hand, institutional pressures develop on construction managers as the expectations of different stakeholder groups drive legitimacy. Combined with limited CSR resources and rationality constraints, they can promote their different division of positioning and form a compatible and complementary relationship chain. These stakeholders share common attributes that enhance their collaborative functioning, such as resource needs, implicit/explicit contracts, internal/external resource flows, and stakeholder attributes. While not directly associated, considering them as 'relationships' for CSR strategy necessitates trust, transparency, and open communication.

Aligned with [80, 81], stakeholders positioning on the map is shaped by their dependency on the company, leading to distinct approaches. Pre-stakeholders, reliant on the company, adopt a coercive approach at the forefront. Meanwhile, post-stakeholders, in symbiotic relations, pursue cooperation, typically at the rear. Coercive approaches hinge on contracts and regulations, while cooperative strategies rely on trust and complementary expertise. Intermediate stakeholders, as shown in Fig 9 –the path of ③, wield substantial decision-making power, often favoring coordinated approaches that demand active engagement. These intermediaries indirectly bridge pre-stakeholders and post-stakeholders, nurturing associated relationships that align with CSR initiatives. The construction company acts as the core, linking these stakeholder groups through various paths, broadening engagement and enhancing legitimacy for compromises resulting from multi-stakeholder involvement. This also implies that different stakeholder groups have different rights, legitimacy and urgency, and display cooperative and proactive strategies to influence corporate activities. Although this mapping often implies one-way communication, research underscores the need for stakeholders to directly influence CSR policies through collaborative efforts. Encouraging collaboration and stakeholder involvement can significantly enhance CSR effectiveness, such as employees partnering with stakeholders to implement environmental standards, promoting more responsible products and services.

## 6. Conclusions

The CSR of an enterprise is a system in which each stakeholder is a component with irreplaceable functions. The relationship between them forms the complementary nature among the stakeholders. The research results show that there is a common relationship among CSR activities, corresponding under CSR and stakeholders, this relationship makes the stakeholders also associated. Chinese construction companies focus on corporate governance and workers interest activities, and then construction quality and environment preservation; and three key stakeholders, namely government, partners and NGOs and media. When companies

implement CSR activities, three types of stakeholders are formed due to the position of stakeholders: pre-stakeholders, intermediate and post-stakeholders; and three stakeholder relationship management approaches are formed correspondingly, i.e., coercive, cooperative and coordinated approaches. The association of stakeholders is the complementarity of stakeholder relationships resulting from the common demands of corporate resources and stakeholders, and can be combined to strengthen the relationships between different stakeholders in order to enhance corporate competitiveness.

This study explores the potential relationships between stakeholders to refine stakeholder-based view and enrich stakeholder interactions. In particular, the proposed associated stakeholder relationship enriches previous stakeholder related research on identification, communication, and dialogue. It helps companies to manage their CSR activities based on stakeholder position and relationships according to the actual needs of CSR, and to improve the efficiency of CSR resources in practice. In addition, because this study is based on a fairer perspective to study the relationship between stakeholders. Equity means that stakeholders receive fair and substantial solutions according to their needs. Future study can continue to further expand from the importance of stakeholders to emphasize the strength and weakness of relationships, refining the relationship between stakeholders under CSR issues, and propose a more complete framework for managing stakeholders.

## Supporting information

**S1 Checklist.**
(DOCX)

**S1 Table. Categorization of CSR in construction.**
(DOCX)

**S2 Table. A list of CSR activities.**
(DOCX)

**S3 Table. ENR's 2022 Top 10 Chinese contractors.**
(DOCX)

**S4 Table. The S-CSR-M.**
(DOCX)

## Author Contributions

**Conceptualization:** Yuqing Zhang, Weiyan Jiang, Xiaowei Wang.

**Data curation:** Yuanshu Liang, Xiaowei Wang.

**Formal analysis:** Yuqing Zhang, Yuanshu Liang.

**Funding acquisition:** Kunhui Ye.

**Investigation:** Yuanshu Liang.

**Methodology:** Yuqing Zhang, Yuanshu Liang, Xiaowei Wang.

**Software:** Yuqing Zhang, Xiaowei Wang.

**Supervision:** Weiyan Jiang, Kunhui Ye.

**Validation:** Kunhui Ye, Xiaowei Wang.

**Visualization:** Yuqing Zhang.

**Writing – original draft:** Yuqing Zhang.

**Writing – review & editing:** Weiyan Jiang.

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
