## [Decision Letter · Decision Letter 0]

11 Sep 2023

PONE-D-23-23102Mapping stakeholders’ relationships management in fulfilling corporate social responsibility: A study of China’s construction industryPLOS ONE

Dear Dr. Ye,

Thank you for submitting your manuscript to PLOS ONE. After careful consideration, we feel that it has merit but does not fully meet PLOS ONE’s publication criteria as it currently stands. Therefore, we invite you to submit a revised version of the manuscript that addresses the points raised during the review process.

Though a reviewer rejected the manuscript, the reviewer provided many valuable and constructive comments. Considering three reviewers’ useful comments and the interesting topic of the manuscript, I would like to give you a chance to revise your manuscript. The revised manuscript will undergo the next round of review by the same reviewers.

We look forward to receiving your revised manuscript.

Kind regards,

Baogui Xin, Ph.D.

Academic Editor

PLOS ONE

Journal Requirements:

 "This research work is supported by the Ministry of Education of the People’s Republic of China (NO.21JHQ092)."

"This research work is supported by the Ministry of Education of the People’s Republic of China (NO.21JHQ092)."

"This research work is supported by the Ministry of Education of the People’s Republic of China (NO.21JHQ092)."

"All authors declare that they have no known competing financial interests or personal relationships that could have appeared to influence the work reported in this paper."

5. Please amend your list of authors on the manuscript to ensure that each author is linked to an affiliation. Authors’ affiliations should reflect the institution where the work was done (if authors moved subsequently, you can also list the new affiliation stating “current affiliation:….” as necessary).

Reviewers' comments:

Reviewer's Responses to Questions

**Comments to the Author**

1. Is the manuscript technically sound, and do the data support the conclusions?

Reviewer #1: Partly

Reviewer #2: Yes

Reviewer #3: Partly

2. Has the statistical analysis been performed appropriately and rigorously? 

Reviewer #1: I Don't Know

Reviewer #2: Yes

Reviewer #3: Yes

3. Have the authors made all data underlying the findings in their manuscript fully available?

Reviewer #1: Yes

Reviewer #2: Yes

Reviewer #3: Yes

4. Is the manuscript presented in an intelligible fashion and written in standard English?

Reviewer #1: No

Reviewer #2: No

Reviewer #3: Yes

5. Review Comments to the Author

Reviewer #1: Ideas are jumbled together in a way that makes it difficult for readers to closely follow along. The literature review, for instance, cites numerous studies, but it is unclear how (many of) the cited studies relate to the present research (in a way that advances arguments or builds theoretical predictions). Despite the coverage of the literature review, which claims are relevant for the present research is unclear. It is my belief that the front-end of the manuscript would need significant re-writing to improve concision, clarity, and structure.

To be more specific, I think the literature review can more directly address these following questions:

(1) Which specific studies motivate an investigation into China's construction industry?

(2) How do CSR initiatives concretely differ based on industries? What attributes lead to these differences?

(3) What do we already know about CSR in the construction industry?

(4) How does the construction industry in China differ from the same industry in different countries? Might these structural differences lead to differences in CSR activities?

(5) What justifies the methodological approach used in the current research? And does this approach differ from other methods used in CSR research? And why is the approach preferable to other potentially viable methods?

Although I am a CSR researcher, I employ different methods than the ones featured in this manuscript. Therefore, I will refrain from making technical comments about the methods as my fellow reviewers may have more relevant comments or concerns.

Reviewer #2: REVIEW REPORT

PONE-D-23-23102

Mapping stakeholders’ relationships management in fulfilling corporate social responsibility: A study of China’s construction industry

I welcome the authors’ efforts to investigate the link between stakeholders’ relationships management and corporate social responsibility. I think the paper is addressing an important issue. However, as you might have expected, I do have some concerns and I would like to explain my concerns and suggestions below, in hopes of helping the authors improve the paper.

1. In my view, authors can also use the lens of legitimacy theory to further support their argument regarding CSR implementation in response of stakeholders’ management/relationship in the context of construction industry. For legitimacy theory, authors can refer the following studies, among others.

• Di Vaio, A., Varriale, L., Di Gregorio, A., & Adomako, S. (2022). Corporate social performance and non‐financial reporting in the cruise industry: Paving the way towards UN Agenda 2030. Corporate Social Responsibility and Environmental Management, 29(6), 1931-1953.

• Velte, P. (2022). Meta-analyses on corporate social responsibility (CSR): a literature review. Management Review Quarterly, 72(3), 627-675.

2. In the literature review section, I would suggest authors to add hypothesis for each link, for instance, regarding stakeholder-based view on CSR and stakeholder relationship management.

3. In the abstract, authors mentioned that “the results show that the SCSR-2M can be composed of three key stakeholders, namely government, partners, NGOs and media.” I think these are four stakeholders not three or authors might consider both NGOs and media as one stakeholder. Confirm the same in rest of the manuscript.

4. Finally, the paper should go through English proof editing. The paper is not very clear to the reader.

Reviewer #3: I only provide suggestions on my own behalf, suggesting accepting this article and also suggesting that the author make some minor modifications.

This article studies stackeholder associations and the research process is clear. A large amount of space was used to describe the data processing process, describing many indicator values, giving readers a strong sense of methodology and technology. However, there is insufficient analysis of the data results and the reasons for their formation, especially in section 5. Findings and Discussion of the paper. The author used nearly 60 literatures to illustrate what? Is it just that the results are consistent with previous research? If so, what is the significance of this article? And the author stated in the abstract that "this article is the earliest effort on this topic". I guess what readers most want to see is the content of this article that is different from existing literatures, and the suggested strategies that are consistent with the data results.

6. PLOS authors have the option to publish the peer review history of their article (what does this mean?). If published, this will include your full peer review and any attached files.

Reviewer #1: No

Reviewer #2: No

Reviewer #3: No

---

## [Author Response · Author response to Decision Letter 0]

20 Oct 2023

Dear editor,

First of all, thank you for giving us a chance to revise the manuscript entitled “Identifying stakeholder clusters in corporate social responsibility fulfillment through an association rule mining approach: A China’s construction study” (ID: PONE-D-23-23102). We revised the manuscript based on the reviewers’ valuable and constructive comments. And, a reply to the reviewers’ comments is included in the file (Revised Manuscript with Track Changes). Based on the comments and suggestions of the reviewers, we made the following revisions: 1) Introducing the basic situation of China’s construction industry as well as the characteristics of CSR. 2) Re-writing Chapter [2. Literature Review], including introducing the legitimacy theory and adding three hypotheses. 3) Reorganize the methodological steps of this study, redraw the flowchart and adjust the relevant parts. 4) Combining Chapter [5. Results and Discussion] with Chapter [2. Literature Review] to further analysis and explain the results of this study. 5) Adjusting the paragraphs of the original manuscript. 6) Proofread the manuscript. Please see manuscript and supplements.

We deeply appreciate your time, and look forward to receiving your feedbacks. If you have any questions, please don’t hesitate to contact us using the address below.

Yours sincerely,

Kunhui Ye

Professor, School of Management Science and Real Estate, Chongqing University

Address: Chongqing University, 83# Shabei Street, Shapingba District, Chongqing, China, 400044

Tel: + 86 023-65120976

E-mail: Kunhui_Ye@Cqu.edu.cn

---

## [Decision Letter · Decision Letter 1]

6 Nov 2023

Mapping stakeholders’ relationships management in fulfilling corporate social responsibility: A study of China’s construction industry

PONE-D-23-23102R1

Dear Dr. Ye,

We’re pleased to inform you that your manuscript has been judged scientifically suitable for publication and will be formally accepted for publication once it meets all outstanding technical requirements.

Kind regards,

Baogui Xin, Ph.D.

Academic Editor

PLOS ONE

Additional Editor Comments (optional):

Reviewers' comments:

Reviewer's Responses to Questions

**Comments to the Author**

1. If the authors have adequately addressed your comments raised in a previous round of review and you feel that this manuscript is now acceptable for publication, you may indicate that here to bypass the “Comments to the Author” section, enter your conflict of interest statement in the “Confidential to Editor” section, and submit your "Accept" recommendation.

Reviewer #1: All comments have been addressed

Reviewer #2: All comments have been addressed

2. Is the manuscript technically sound, and do the data support the conclusions?

Reviewer #1: Yes

Reviewer #2: Yes

3. Has the statistical analysis been performed appropriately and rigorously? 

Reviewer #1: I Don't Know

Reviewer #2: Yes

4. Have the authors made all data underlying the findings in their manuscript fully available?

Reviewer #1: Yes

Reviewer #2: (No Response)

5. Is the manuscript presented in an intelligible fashion and written in standard English?

Reviewer #1: Yes

Reviewer #2: (No Response)

6. Review Comments to the Author

Reviewer #1: I appreciate the thoroughness in the authors' response to the concerns that I raised in the previous review. I believe that the authors have sufficiently addressed my comments in this revision round.

Reviewer #2: (No Response)

7. PLOS authors have the option to publish the peer review history of their article (what does this mean?). If published, this will include your full peer review and any attached files.

Reviewer #1: No

Reviewer #2: No

---

## [Editor Report · Acceptance letter]

26 Dec 2023

PONE-D-23-23102R1 

PLOS ONE

Dear Dr. Ye, 

I'm pleased to inform you that your manuscript has been deemed suitable for publication in PLOS ONE. Congratulations! Your manuscript is now being handed over to our production team.

Kind regards, 

on behalf of

Professor Baogui Xin 

Academic Editor

PLOS ONE